Diuron tolerance and potential degradation by pelagic microbiomes in the Great Barrier Reef lagoon

Angly Florent E. 1 florent.angly@gmail.com
Pantos Olga 1 2
Morgan Thomas C. 1
Rich Virginia 3 4
Tonin Hemerson 5
Bourne David G. 5
Mercurio Philip 5 6
Negri Andrew P. 5
Tyson Gene W. 1
1 Australian Centre for Ecogenomics, The University of Queensland , St Lucia, Queensland , Australia
2 Global Change Institute, The University of Queensland , St Lucia, Queensland , Australia
3 Department of Soil, Water and Environmental Science, The University of Arizona , Tucson, AZ , United States of America
4 Microbiology Department, The Ohio State University , Columbus, OH , United States of America
5 Australian Institute of Marine Science , Townsville, Queensland , Australia
6 National Research Centre for Environmental Toxicology, The University of Queensland , Coopers Plains, Queensland , Australia
Smidt Hauke
Electronic publication date: 2016 Mar 8
Publication date: 2016
Volume: 4
Electronic Location ID: e1758
Received 2015 May 19; Accepted 2016 Feb 13
Copyright: ©2016 Angly et al.
Copyright year: 2016
Copyright holder: Angly et al.
License: This is an open access article distributed under the terms of the Creative Commons Attribution License, which permits unrestricted use, distribution, reproduction and adaptation in any medium and for any purpose provided that it is properly attributed. For attribution, the original author(s), title, publication source (PeerJ) and either DOI or URL of the article must be cited.
License URL: https://creativecommons.org/licenses/by/4.0/

Keywords: Diuron, Great barrier reef, Amplicon profiling, Metagenomics, Herbicide, Incubation

Funding: Australian Research Council’s Discovery Early Career Research Award DE120101213 Queen Elizabeth II fellowship DP1093175 Australian Government’s National Environmental Research Program Tropical Ecosystems Hub Project 4.2 This study was supported by the Australian Research Council’s Discovery Early Career Research Award to Florent Angly (DE120101213) and Queen Elizabeth II fellowship to Gene Tyson (DP1093175) and the Australian Government’s National Environmental Research Program Tropical Ecosystems Hub Project 4.2. The funders had no role in study design, data collection and analysis, decision to publish, or preparation of the manuscript.

==============================
Diuron is a herbicide commonly used in agricultural areas where excess application causes it to leach into rivers, reach sensitive marine environments like the Great Barrier Reef (GBR) lagoon and pose risks to marine life. To investigate the impact of diuron on whole prokaryotic communities that underpin the marine food web and are integral to coral reef health, GBR lagoon water was incubated with diuron at environmentally-relevant concentration (8 µg/L), and sequenced at specific time points over the following year. 16S rRNA gene amplicon profiling revealed no significant short- or long-term effect of diuron on microbiome structure. The relative abundance of prokaryotic phototrophs was not significantly altered by diuron, which suggests that they were largely tolerant at this concentration. Assembly of a metagenome derived from waters sampled at a similar location in the GBR lagoon did not reveal the presence of mutations in the cyanobacterial photosystem that could explain diuron tolerance. However, resident phages displayed several variants of this gene and could potentially play a role in tolerance acquisition. Slow biodegradation of diuron was reported in the incubation flasks, but no correlation with the relative abundance of heterotrophs was evident. Analysis of metagenomic reads supports the hypothesis that previously uncharacterized hydrolases carried by low-abundance species may mediate herbicide degradation in the GBR lagoon. Overall, this study offers evidence that pelagic phototrophs of the GBR lagoon may be more tolerant of diuron than other tropical organisms, and that heterotrophs in the microbial seed bank may have the potential to degrade diuron and alleviate local anthropogenic stresses to inshore GBR ecosystems.

Introduction

Coral reefs are very rich and diverse ecosystems, though due to both local and global anthropogenic disturbances, they are in a state of gradual decline (Pandolfi et al., 2003). Local impacts such as coastal pollution (Brodie et al., 2012) and overfishing (De’ath et al., 2012) affect both the reef macrobiota (Fabricius, 2005; Sandin et al., 2008) and the microorganisms in their associated microbiomes (Dinsdale et al., 2008; Thurber et al., 2009; Webster et al., 2011). Microorganisms not only underpin the marine food web and system function (Azam et al., 1983; Fenchel, 2008), but also form intimate relationships with corals that are essential for their health (Rohwer et al., 2002; Lesser et al., 2004; Lema, Willis & Bourne, 2012). Hence, changes to microbiome structure and function can compromise the health of coral reefs and their ability to recover from stresses (Ainsworth, Thurber & Gates, 2010; Hughes et al., 2010).

Fishing impact is limited across the Great Barrier Reef (GBR), but its otherwise oligotrophic inshore coastal habitats (Schaffelke et al., 2012) are subject to pollution from land runoff of agricultural, industrial and domestic origin (Packett et al., 2009; Brodie et al., 2012). For example, herbicides that are used to control weeds in the sugarcane plantations of Queensland have been detected in the waterways of the catchments (McMahon et al., 2005; Mitchell, Brodie & White, 2005; Shaw & Müller, 2005), intertidal sediments, and in the waters surrounding inshore coral reefs (Shaw & Müller, 2005; Lewis et al., 2009). The herbicide diuron, or 3-(3,4-dichlorophenyl)-1,1-dimethylurea (DCMU), is consistently detected in the GBR lagoon (Haynes, Müller & Carter, 2000; Shaw et al., 2010; Kennedy et al., 2012b), exceeding the Australian and New Zealand guideline trigger value of 0.2 µg/L at some sites (Smith et al., 2012). Diuron is an effective broad-spectrum herbicide due to its ability to inhibit the photosystem II (PSII) complex of photosynthetic organisms (Metz et al., 1986). Consequently, it poses risks to a wide range of marine eukaryotes including invertebrates (Bellas et al., 2005; Mai et al., 2013), seagrasses (Haynes et al., 2000; Flores et al., 2013), fishes (Mhadhbi & Beiras, 2012), diatoms (Legrand et al., 2006; Magnusson, Heimann & Negri, 2008) and microalgae, both benthic (Magnusson et al., 2012) and in endosymbiotic relationship with corals (Jones et al., 2003; Jones, 2004; Shaw, Brodie & Müller, 2012). Furthermore, diuron’s low rate of abiotic degradation by hydrolysis and photo-degradation (Okamura, 2002; Moncada, 2004; Mercurio et al., 2015) results in its accumulation in the marine environment, particularly in sediments (Haynes, Müller & Carter, 2000; Balakrishnan, Takeda & Sakugawa, 2012; Xu et al., 2013).

In addition to its effects on eukaryotes, diuron is also toxic to bacteria. Low concentrations of diuron (1.6–23 µg/L) impairs the photosynthesis of phototrophs such as cyanobacteria (Allen et al., 1983; Deng, Gao & Sun, 2012), while very high concentrations (1 × 106 µg/L) inhibit the growth of bacterial heterotrophs (Faÿ et al., 2010). Consequently, exposure to a pulse of diuron through a flooding event can significantly alter microbiome structure (Tlili et al., 2008) and decrease microbial abundance (Ricart et al., 2009). Conversely, diuron exposure can also increase bacterial abundance in wastewater treatment ponds (Sumpono et al., 2003), likely due to heterotrophic bacteria taking advantage of the release of organic compounds by organisms susceptible to diuron, such as diatoms (Proia et al., 2011). An alternative explanation is that some bacteria are able to metabolize diuron and use it as an energy source (Dellamatrice & Monteiro, 2004), as seen in soil, sediments and sludge (Cullington & Walker, 1999; Dellamatrice & Monteiro, 2004; Sørensen, Albers & Aamand, 2008; Stasinakis et al., 2009; Pesce et al., 2012). This biodegradation is catalyzed by phenylurea hydrolases (PuhAB) (Turnbull et al., 2001; Khurana et al., 2009) and proceeds faster than degradation by abiotic means (Cullington & Walker, 1999; Dellamatrice & Monteiro, 2004; Sørensen, Albers & Aamand, 2008; Stasinakis et al., 2009; Pesce et al., 2010). Microorganisms are therefore regularly employed in herbicide pollution remediation technologies (Villaverde et al., 2012; Safi, Awad & El-Nahhal, 2014).

A recent study found that microbial community variations in the GBR lagoon are primarily driven by riverine effluents (Angly et al., 2016) and a year-long seawater simulation experiment presented evidence that microorganisms play a role in the biodegradation of a wide range of PSII herbicides in this region (Mercurio et al., 2015). Although the community-wide effects of diuron on marine microorganisms are not characterized, this research suggests that pelagic microbiomes may protect coral reef and seagrass ecosystems by degrading this herbicide. In the present study, we hypothesized that diuron significantly affects the composition of these microbiomes by specifically: (i) inhibiting phototrophs in the short-term; and (ii) sustaining the long-term growth of selected heterotrophs that have the potential to metabolize it, leading to the herbicide’s disappearance. To investigate these hypotheses, we collected samples from the simulation study of Mercurio et al. (2015) and an inshore GBR location, and characterized their microbiomes (Archaea and Bacteria) using high-throughput 16S rRNA amplicon profiling and metagenomics.

Materials & Methods

Cape Ferguson diuron incubations

Mercurio et al. (2015) collected surface seawater (24 L) in sterile containers in the tropical dry season (15 May 2012) from Cape Ferguson, QLD, Australia (latitude −19.2673297, longitude 147.0591537) (Fig. S1), a site that is ∼17 km downstream from the Haughton River mouth and where diuron is consistently reported in the wet season (Lewis et al., 2009; Kennedy et al., 2012a; Kennedy et al., 2012b). The investigators passed seawater through 20 µm impact filters, dispensed it in 500 mL glass flasks and incubated it during 365 days on a shaking platform at 25 °C either in the dark or in the light (12:12 light day cycle with 40 µmol photons m−2 s−1), and with or without amendment of diuron (at the ecologically relevant concentration of 8 µg/L) (Lewis et al., 2009; Kennedy et al., 2012a; Kennedy et al., 2012b) (Figs. S2A and S2B). The investigators performed each experimental treatment in triplicate (12 flasks in total) and monitored diuron concentration for each flask over the life of the incubation. For more details of the experimental set up and diuron degradation results, see Mercurio et al. (2015).

In the present study, we collected subsamples (705 µL) from each flask of the Cape Ferguson incubation experiment at day 0, 2, 7, 28, 120 and 365 for 16S rRNA gene analysis. Each subsample was collected with a pipette after thoroughly shaking the flask, 5 µL were stained with 5 µL of DAPI (1 µg/mL) in the dark for 15 min, and observed with a Nikon Ci-L epifluorescence microscope (Fig. S2C). This confirmed the presence of DNA-containing cells, a prerequisite for sequencing.

Total DNA was extracted by first centrifuging each sample at 13,000 × g for 30 min. Each resulting pellet was then resuspended in 20 µL microLYSIS-Plus DNA release buffer (Microzone, West Sussex, UK) for 30 min at room temperature and incubated using a thermal cycler following the manufacturer’s tough cell lysis protocol (65 °C for 15 min; 96 °C for 2 min; 65 °C for 4 min; 96 °C for 1 min; 65 °C for 1 min; 96 °C for 30 s). Control of the absence of microLYSIS-Plus reagent contamination by foreign DNA was carried out by adding a blank sample, containing only the microLYSIS-Plus buffer (without template DNA).

Microbial amplicons were generated by PCR-amplifying the V6–V8 variable regions of the 16S rRNA gene in the total DNA using a universal primer set targeting Archaea and Bacteria (iTAG926F and iTAG1392wR primers) according to the protocol in Dove et al. (2013). These amplicons were paired-end sequenced on an Illumina MiSeq instrument at the Australian Centre for Ecogenomics (ACE).

Microbiome profiling

Amplicon reads were processed using Hitman (https://github.com/fangly/hitman, described in Angly et al., 2016), a bioinformatic workflow based around the UPARSE methodology (Edgar, 2013). The following parameters were used: trimming length of 250 bp, minimum quality value of 7 (16 for HiFi sequences), maximum number of expected errors of 3.0 (0.5 for HiFi sequences), OTU clustering at 97% identity (species-level), GOLD (Bernal, Ear & Kyrpides, 2001) as the reference database for chimera detection, rarefaction depth of 20,000 counts, minimum global alignment of 95% identity (genus-level) for taxonomic annotation using the merged Silva (Quast et al., 2012) and Greengenes (McDonald et al., 2012) databases (https://github.com/fangly/merge_gg_silva), gene-copy number correction with CopyRighter (Angly et al., 2014), and exclusion of taxa matching “Eukaryota*” or “*Chloroplast*.”

Rarefaction curves were produced using Bio-Community’s bc_accumulate (Angly, Fields & Tyson, 2014) with 100 random subsets. After taxonomic assignments and gene-copy number correction within Hitman, calculation of α-diversity was performed using Bio-Community bc_measure_alpha at the selected rarefaction depth. A few OTUs of interest, that could not be taxonomically assigned with Hitman, were classified by the RDP Classifier at 80% confidence (Cole et al., 2009), and Silva’s SINA with 95% identity (Quast et al., 2012).

Statistical analysis

The significance of changes in taxon relative abundance between sampling points were evaluated using LEfSe (Segata et al., 2011). The effects of incubation regimen on microbial community structure was assessed by Hellinger-transforming the microbial profiles and analyzing them using the R language (R Foundation for Statistical Computing, Vienna, Austria, 0000), specifically using the capscale(), adonis() and rda() functions of the vegan packages (Dixon, 2003), for PCoA, PERMANOVA and RDA analysis respectively.

Dunk Island metagenome preparation

An additional seawater sample was collected in the tropical dry season (13 October 2009), north of Dunk Island, QLD, Australia (latitude −17.9242918, longitude 146.1429637) (Fig. S1). This site is ∼15 km downstream from the Tully River mouth, and exposed to diuron and other PSII herbicides in comparable concentrations as the Cape Ferguson site (Lewis et al., 2009; Kennedy et al., 2012a; Kennedy et al., 2012b). A 20 L volume was taken from a depth of 5 m and pre-filtered through a 2.7 µm Whatman GF/D filter and a 1.6 µm Whatman GF/A filter to remove particles and most eukaryotic microorganisms. The filtrate was then passed through a 0.22 µm Millipore Express Plus filter to capture the bacterial and archaeal fraction. The filters were folded in half, cells inward, added to a tube containing 20 mL of lysis buffer (40 mM Na2 EDTA, 50 mM Tris pH 8.3 and 0.73 M sucrose, sterilized), stored shipboard at −20 °C and transferred to −80 °C on land.

DNA was extracted from the filter using a modified method from Suzuki et al. (2004). In brief, the filter was thawed on ice, added 6 mL of lysis buffer with 5 mg/mL lysozyme and the tube was incubated for 30 min at 37 °C, while rotating at 10 rpm. Proteinase K (1.1 mg/mL final concentration) and 10% sodium dodecyl sulfate (1.1% final concentration) were added and the sample was incubated at 55 °C for 2 h, with rotation. The lysate was split in half and DNA was extracted from each using two rounds of phenol:chloroform:isoamyl alcohol (25:24:1, pH 8.0), then one round of chloroform:isoamyl alcohol (24:1). Aqueous phases were pooled and frozen overnight at −20 °C. The aqueous phase was then cleaned by passage of 15 mL at a time through Amicon Ultra-15 100 kDa spin unit (EMD Millipore, Billerica, MA, USA). The filter was washed once with 8 mL of Tris EDTA buffer (TE, 10 mM, pH 8.0) and recovered with 50 µL of TE (1 mM, pH 8.0). DNA was then further cleaned by precipitation with 70% ethanol, the pellet was washed once with 70% ethanol, air dried, and resuspended in 100 µL TE, for a total yield of 37 µg DNA. The resulting DNA was sequenced on an Illumina (Solexa) Genome Analyzer II instrument at the University of Arizona, producing 25.4 million pairs of 101 bp long reads.

Read-centric metagenomic screening for phenylurea hydrolases

The Dunk Island metagenomic read pairs were cleaned by removing Illumina adapters with TRIMMOMATIC, merged using PEAR (but keeping unmerged read pairs), 5′ end quality-trimmed at the first nucleotide below Q13 and filtered to remove sequences smaller than 60 bp using TRIMMOMATIC. The resulting quality-controlled reads were compared to all known PuhAB phenylurea hydrolase proteins (GI 218764925, 598062302and 218764905), belonging to the metal-dependent amidohydrolase superfamily (Turnbull et al., 2001; Khurana et al., 2009), using BLASTX (Camacho et al., 2009). The BLAST database also included 55 other closely-related proteins, including other herbicide hydrolases, to ensure the specificity of the results: the MolA molinate hydrolase (Sugrue et al., 2015) (GenBank FN985594), four LibA linuron hydrolases (Bers et al., 2011; Bers et al., 2013) (GenBank JN104629, JN104630, JN104631and JN104633) and 50 proteins from the metal dependent amidohydrolase superfamily (GI 18655481, 7245484, 23200144, 23200220, 3892028, 22218649, 14719683, 13786715, 28948588, 30749918, 999767, 24987382, 27574194, 30750126, 24371617, 40787177, 15966345, 16124371, 5817646, 22972062, 21222419, 23058081, 24216335, 3912984, 1709955, 33595951, 27375360, 27378941, 22987263, 23105179, 3914514, 16763233, 27377792, 2829648, 18311855, 15612748, 17540282, 17548772, 38108196, 15791459, 15528804, 40063581and 24371695). Significant similarities (E-value < 1e-6) were extracted and their alignment to the most similar proteins was visually inspected using Jalview (Waterhouse et al., 2009). The putative taxonomic affiliation of the Puh-like proteins was established by comparing the metagenomic reads to the NCBI nt database using TBLASTX.

Under the assumption that each distinct identified PuhAB protein is encoded by a different species (the three known Puh proteins are encoded on three distinct genomes), we approximated this species’ relative abundance as: A ≈ R × G × 10−4/(P × M × L)%, where R is the number of reads matching puhAB, M is the number of screened metagenomic reads (22,927,633), L is the average read length L (93.7 bp), P is the average length of the puhAB genes (1,376 bp), G is the average genome length in marine microbiomes (2.58 Mbp) (Angly et al., 2009), and S is the number of species in inshore GBR water column (643 OTUs in the diuron incubation experiment).

Contig-centric metagenomic screening for photosystem genes

For this analysis, the Dunk Island metagenomic raw read pairs were cleaned with TRIMMOMATIC by removing Illumina adapters, deleting reads with uncalled bases, truncating their 5′ end to a final length of 80 bp, and removing smaller reads. The data were assembled using IDBA-UD (Peng et al., 2012), and the resulting scaffolds translated into their six possible reading frames. The hmmsearch tool of HMMER3 (Eddy, 2011) was employed to look for photosystem B proteins in these translated scaffolds using the TIGR001151 PsbA hidden Markov Model profile of TIGRFAMs (Haft, Selengut & White, 2003). A maximum E-value of 1e-50 was used to retrieve significant matches and their alignment was visualized in Jalview. The taxonomic affiliation of the scaffolds matching PsbA was determined by best BLASTN similarity against the NCBI nr database (minimum identity of 70% over a minimum alignment length of 1,200 bp, i.e., the length of PsbA + 40 amino acids). Nesoni (https://github.com/Victorian-Bioinformatics-Consortium/nesoni) and SHRiMP (Rumble et al., 2009) were used to map the Illumina reads against the metagenomic scaffolds and call single nucleotide polymorphisms (SNPs).

Results & Discussion

Microbial dynamics in Cape Ferguson diuron incubation

Mercurio et al. (2015) collected seawater during the tropical dry season at Cape Ferguson, an inshore region of the GBR (Figs. S1 and S2), to conduct a year-long diuron incubation experiment. In the present study, the 16S rRNA amplicon sequencing of 72 samples taken at set time points from the incubation flasks generated a total of 4.96 million read pairs (NCBI accession PRJNA276057). Processing through the Hitman bioinformatic pipeline resulted in 3.83 million high-quality sequences (77.3% of the initial amount). Rarefaction at a depth of 20,000 counts per sample provided a sequencing depth-independent view of the diversity of the samples (Fig. S3, Table S1), collectively containing 4,743 distinct OTUs (97% identity level).

Figure 1 Heatmap showing the relative abundance of microbial genera over the one-year Cape Ferguson diuron incubations.

The four incubation conditions are control + dark (C_D), control + light (C_L), diuron + dark (D_D) and diuron + light (D_L). The three replicates of each incubation condition were averaged and only microbial genera reaching 1% are indicated.

The taxonomic affiliation conducted by best global alignment against the Greengenes database and subsequent gene-copy number correction (Angly et al., 2014) permitted estimation of changes in the relative abundance of prokaryotic taxa over time (Fig. S4). When averaging the replicates (Fig. 1, Fig. S5), the most abundant taxa at the start of the incubation (day 0) included the orders Rickettsiales (19%) and Synechococcales (14% average relative abundance), from the Cyanobacteria and Proteobacteria phyla, respectively. The microbiomes were marked by a succession of various taxa over time, as seen in previous work (Fierer et al., 2010). For example, Sphingomonadales increased significantly from an initial average of 0.38% relative abundance (day 0) to dominate the communities with 30% at day 2 (LEfSe; α < 0.05). Rhodobacterales-affiliated sequences increased significantly, reaching a maximum relative abundance of 36% on average a week after the start of the incubation (day 7) (LEfSe; α < 0.05), and subsequently significantly decreased until day 120 (LEfSe; α < 0.05). At the end of the incubation experiment (day 365), Oceanospirillales were very abundant in the control flasks exposed to light, while Thiotrichales dominated the samples incubated in the dark (both control and diuron-treated).

Three predominant OTUs (OTU 12, 13 and 20) characteristic of the flasks kept in the dark could not be assigned to a taxonomic group. Further identification efforts using the RDP Classifier and Silva’s SINA suggest that they all belong to the Proteobacteria phylum, more precisely to the Salinisphaera, Coxiella and GR-WP33-30 taxa (Table S2). The genus Salinisphaera includes a recently sequenced species that is adapted to environments with fluctuating conditions (Antunes et al., 2011), while the genus Coxiella contains a single species that is highly resistant to environmental stresses such as temperature, osmotic pressure and ultraviolet radiation (Voth & Heinzen, 2007), and representatives of the order GR-WP33-30 were detected in uranium mines (Selenska-Pobell & Radeva, 2004). The robustness of these taxa may be responsible for their success in the dark and likely oligotrophic conditions of the incubation flasks.

Effect of diuron on microbial profiles

The diuron measurements made by Mercurio et al. (2015) in the incubation flasks ranged from an initial 8.77 µg/L (dark conditions, replicate R4) down to 3.78 µg/L (light conditions, replicate R3) after one year of incubation. Here, we included the diuron concentration of each individual flask as an input for a constrained ordination (Fig. 2), which demonstrated a significant influence of incubation time and light exposure, but not of diuron concentrations on the microbial profiles (PERMANOVA, p < 0.05). Dissection of the differences between diuron-treated and control flasks for each individual sampling day, PCoA (Fig. S6) confirmed that diuron did not affect microbiome composition significantly (PERMANOVA, p < 0.05).

Figure 2 OTU-level RDA of the microbiomes (Hellinger-based) in the Cape Ferguson diuron incubations.

OTUs are indicated by a red cross and the Greengenes taxonomic affiliation of the most discriminating is shown. Circle size is proportional to incubation time (sampling day). Environmental factors are green arrows depicting light amount in the light and dark treatments, incubation time and measured diuron concentration. Asterisks denote environmental factors that are statistically significant (PERMANOVA; p < 0.05). Samples from day 2 and 7 were omitted from this analysis because diuron concentration was not measured on these days.

Resistance of photosynthetic bacteria to diuron

Some phototrophic bacteria are inhibited by diuron, while others are insensitive. For example, photosystems I and II exist in Cyanobacteria and vascular plants, and physiological experiments have demonstrated binding of diuron on the cyanobacterial photosystem II, leading to photosynthesis inhibition (Allen et al., 1983; Gadkari, 1988; Brusslan & Haselkorn, 1989; Deng, Gao & Sun, 2012). Conversely, diuron does not bind to the photosynthetic reaction center of purple bacteria and they may remain unaffected (Sinning, 1992). We thus hypothesized that the majority of phototrophic prokaryotes in the incubation experiment would be affected by diuron toxicity, resulting in their rapid decline.

Cyanobacteria and purple bacteria such as Rhodobacteraceae were prevalent in the incubation flasks, but their relative abundance did not decline between day 0 and 28 (Fig. 2, Figs. S6A–S6D), despite the presence of more than 8.45 µg/L diuron on average during this period, a concentration that markedly inhibits the photosynthesis of diatoms and green algae (Magnusson, Heimann & Negri, 2008; Magnusson et al., 2012). This supports previous reports that Cyanobacteria are less sensitive to PSII herbicides than eukaryotic phototrophs (Lürling & Roessink, 2006). The relative insensitivity of Cyanobacteria in our dataset could be explained by pollution-induced community tolerance (PICT) following chronic exposure to herbicides, which was previously reported for biofilms in a French river (Tlili et al., 2008; Tlili et al., 2011) and for periphyton (a mixture of detritus, algae and microorganisms growing on submerged surfaces) in the GBR lagoon (Magnusson et al., 2012) and in a Swedish fjord (Molander & Blanck, 1992). The mechanism underpinning this tolerance for diuron is not yet elucidated, but may be related to the evolution and enrichment of high-turnover variants of the PsbA protein upon which diuron and other PSII herbicides such as irgarol 1051 bind (Eriksson et al., 2009; Deng, Gao & Sun, 2012).

Metagenomic analysis was undertaken to explore the presence of PsbA variants and the potential for diuron resistance in the GBR lagoon. Since the samples collected during the Cape Ferguson incubation experiment contained too little biomass for comprehensive metagenomic sequencing, we prepared a metagenome from a sample collected during the tropical dry season at Dunk Island, another inshore GBR location (NCBI accession SRR1819825). Weather, river effluent and diuron exposure data indicate that the Dunk Island and Cape Ferguson samples were both representative of the GBR lagoon during the dry season, when the effects of riverine floodwaters are minimal (Text S1, Tables S3–S6), and therefore comparable (Angly et al., 2016). The Dunk Island metagenome was assembled into ∾74,000 scaffolds (771 bp average length, 879 bp N50) from which all putative PsbA protein sequences were identified (Fig. 3). The introduction of Val219 and Ser264 mutations in PsbA confers PSII herbicide resistance (Bettini et al., 1987; Mengistu et al., 2000) and mutations in the PsbA PEST domain (rich in amino-acids P, E, S and T) were previously correlated with resistance in the environment (Eriksson et al., 2009). But none of these mutations were detected in the Dunk Island scaffolds of Cyanobacteria (Synechococcus and Prochlorococcus). Further, a total of ∼24,900 reads mapped onto the 12.9 kb long Prochlorococcus scaffold, but no SNPs could be identified within the psbA gene.

Figure 3 Sequence alignment of the PsbA proteins predicted from Dunk Island metagenomic scaffolds.

Residues are colored based on the Clustal X scheme. Locations previously correlated with herbicide resistance are indicated by a box. The bold sequence corresponds to PEST type 11, implicated in resistance against the irgarol PSII herbicide. The bottom panel represents the number of conserved amino acids at each position and their consensus and the column on the right the BLASTN taxonomic classification of the scaffolds.

PEST sequence type 11 (RETTENESANAGYK), representing a PEST type hypothesized to confer irgarol tolerance to the Swedish fjord microbiomes (Eriksson et al., 2009), was detected in the metagenomic scaffolds (Fig. 3). However, BLASTN analyses suggest that this sequence was part of a eukaryotic genome (100% query coverage and 100% identity to Micromonas, a member of the Prasinophyceae class) (Worden et al., 2009). The Prasinophyceae are composed of unicellular photosynthetic green algae and Micromonas (<2 µm) could have passed through the 1.6 µm wide pores of the filters used during metagenome preparation. Similarly, the Swedish fjord sequencing read containing PEST type 11 (accession AM933747) best matched a eukaryotic genome from another subdivision of the Prasinophyceae (100% query coverage and 99% identity to the Pycnococcaceae family). This suggests that PEST type 11 is a general feature of the Prasinophyceae genome and, at least in the present study, not an adaptive mutation of Bacteria and Archaea to protect against PSII herbicides.

BLASTN investigation of the metagenomic scaffolds from Dunk Island encoding PsbA revealed that they were not only prokaryotic and eukaryotic. Most of them (14 out of 24) were of viral origin (Fig. 3), and 9 out of the 11 PEST types identified in the present study matched some carried by cyanophages. This large PEST type diversity and the propensity of phages to transfer genes to and from their hosts (lateral gene transfer) raises the possibility that cyanophages steer the stability of PsbA in their hosts (Lindell et al., 2004; Zeidner et al., 2005; Sharon et al., 2007). Future research should consider how this may influence the tolerance of phototrophs to PSII herbicides.

Potential for degradation by heterotrophic bacteria

Another hypothesis formulated in this study was that specific heterotrophic populations would carry genes for the degradation of diuron and take advantage of this resource, leading to their increase in relative abundance over time. During the one-year Cape Ferguson incubation experiment, Mercurio et al. (2015) reported 15–31% diuron degradation. They attributed this slow degradation in part to prokaryotic breakdown but, in the present work, we found no significant association between diuron-treated incubation flasks and heterotrophic abundance between day 28 and 365 (Fig. 2, Figs. S6D–S6F). While this evidence goes against our hypothesis of rapid heterotrophic degradation, a similar marine incubation study also detected a lack of diuron degradation over a shorter 42 d timeframe (Thomas, McHugh & Waldock, 2002).

Experiments using microbiomes from soil (Attaway, Paynter & Camper, 1982; Cullington & Walker, 1999; Widehem et al., 2002; Dellamatrice & Monteiro, 2004; Ngigi et al., 2011), activated sludge (Stasinakis et al., 2009) and freshwater sediments (Ellis & Camper, 1982; Pesce et al., 2010) have demonstrated that diuron can be degraded by bacteria belonging to the genera Pseudomonas (El-Deeb et al., 2000), Arthrobacter (Turnbull et al., 2001; Villaverde et al., 2012), Mycobacterium (Khurana et al., 2009), Variovorax (Sørensen, Albers & Aamand, 2008), Bacillus, Vagococcus and Burkholderia (Ngigi et al., 2011). Sequences affiliated with some of these taxa, specifically Burkholderia and Pseudomonas, were detected at ∼4% in our marine incubations, but their relative abundance did not change significantly in response to long-term exposure to this herbicide (Fig. 2, Fig. S1), suggesting that exposure to diuron and any potential degradation did not alter their evolutionary fitness.

To explore the reasons for the lack of rapid heterotrophic degradation of diuron in the incubation experiment and the potential for diuron degradation in the GBR lagoon at large, we looked for phenylurea hydrolase genes, known to degrade diuron (Turnbull et al., 2001; Khurana et al., 2009), in the reads of the Dunk Island metagenome. Two reads had BLASTX similarities to a protein database covering the metal-dependent amidohydrolase superfamily. These reads were more similar to PuhB than to other PSII hydrolases and proteins from the same superfamily, with a high 51% amino acid identity over 54 amino acids (Fig. S7) indicative of the presence of potential phenylurea hydrolase homologs (Rost, 1999). Despite the short length of these metagenomic reads and the potential sequencing errors they contain, the proteins identified here may represent novel phenylurea or other PSII hydrolases, whose existence has previously been suggested (Pesce et al., 2012). Further research will be needed to characterize the sequence, structure and function of this protein and thus confirm this hypothesis. The five top scoring similarities of one of these metagenomic reads (TBLASTX, ≥98% query cover, E value ≤ 2e-22) suggest that a bacterium from the Bacteroidetes phylum (Flavobacteriia or Cytophagia order) encodes this PuhB-like protein, while the five top scoring similarities for the other read (TBLASTX, ≥97% identity, E value ≤ 5e-12) did not agree on a precise taxonomic origin. Further, calculations (see ‘Materials & Methods’ section) indicate that this putative hydrolase could be present in low-abundance species, in the tail of the microbial rank-abundance curve (∼0.01% relative abundance). Overall, these findings suggest that the marine microbial seed bank, “a reservoir of dormant individuals that can potentially be resuscitated” (Lennon & Jones, 2011), may have a potential for herbicide degradation.

Microbial enrichment studies that reported rapid biodegradation of diuron were conducted with a rich substrate or supplemented with alternative sources of carbon and nitrogen, sometimes under the form of soil or sediments (Widehem et al., 2002; Sørensen, Albers & Aamand, 2008). From this evidence, we conclude that resources may be a limiting factor for marine heterotrophs to express their diuron-degrading potential in often oligotrophic marine waters (Schaffelke et al., 2012), as is the case in marine incubations performed without supplementation (Thomas, McHugh & Waldock, 2002; Mercurio et al., 2015). GBR microorganisms are thought to metabolize nutrients from land runoff at inshore sites (Alongi & McKinnon, 2005) and, given that these sites receive high diuron and nutrient input during the wet season (Packett et al., 2009) perhaps along with diuron-degrading species, we predict that heterotrophic diuron degradation may be enhanced episodically in the GBR lagoon.

Conclusions

This study used amplicon and metagenomic sequencing to evaluate the effects of a PSII herbicide on the composition of entire prokaryotic communities, rather than selected species. It provides a baseline for future research on the impacts of herbicides on the marine ecosystem by suggesting that the effects of the PSII herbicide diuron on communities of GBR near-shore pelagic prokaryotes are limited. Metagenomic evidence suggests that prokaryotic heterotrophs in the marine water column may encode potential new herbicide hydrolase genes, though their expression may be limited by scarce environmental resources in the dry season. The apparent tolerance of marine pelagic phototrophs to diuron may have been due to the acquisition of a resistance mechanism following regular exposure to this herbicide. While no PEST sequence mutations in Cyanobacteria could explain this resistance in the present study, resident phages carried various PEST sequence types and could act as a reservoir. In summary, many components of coral reef ecosystems are stressed by herbicides from land runoff, but in contrast, the pelagic microbiome that underpins the marine food web and is integral to reef functioning, may represents an important buffer that mitigates the impacts of local anthropogenic and natural stresses on coral reefs.

Supplemental Information

Figure S1 Map of the sampling locations

The sampling locations within Australia (inset) and the Great Barrier Reef lagoon (main panel) are shown. The main rivers influencing these sites are depicted in blue and the arrow indicates the direction of the main current.

Click here for additional data file.

Figure S2 Details of the Cape Ferguson diuron incubation experiment

(A) Triplicate flasks incubated at 25 °C with or without diuron amendment, and with or without light for 365 days. (B) Example of a flask incubated with diuron in the light and in which signs of growth are visible at day 365. (C) Micrograph showing DAPI-stained cells in a diuron-treated light-incubated flask collected at day 150.

Click here for additional data file.

Figure S3 Rarefaction curves for the Cape Ferguson incubations

These rarefaction curves show sample OTU diversity as a function of sequencing depth for each experimental treatments: (A) Chao1 richness and (B) Shannon-Wiener index. The black dashed line indicates the rarefaction depth used in this study.

Click here for additional data file.

Figure S4 Heatmap of microbial genera relative abundance over the one-year Cape Ferguson incubations

The four incubation conditions are control + dark (C_D), control + light (C_L), diuron + dark (D_D) and diuron + light (D_L), with each replicate shown (R2, R3 and R3). Only microbial genera reaching 1% are indicated.

Click here for additional data file.

Figure S5 Dynamics of the microbiomes over the one-year Cape Ferguson incubations

The three replicates in each of the four incubation conditions were averaged and only microbial orders reaching 10% are indicated.

Click here for additional data file.

Figure S6 OTU-level PCoA of the Cape Ferguson microbiomes (Hellinger-based) for all incubation times

Circle size is proportional to sampling day. The p-values from PERMANOVA tests of the differences between diuron-treated and control incubations are shown. Red crosses indicate OTUs that drive sample differences.

Click here for additional data file.

Figure S7 Alignment of two metagenomic reads from Dunk Island against known phenylurea hydrolases (position 365-445)

Residues are colored based on the Clustal X scheme. The bottom panel represents the number of conserved amino acids at each position and their consensus.

Click here for additional data file.

Table S1 Microbial diversity in the Cape Ferguson incubation flasks

The number of OTUs, species richness (Chao1) and overall species diversity (Shannon-Wiener index) were calculated after rarefaction, taxonomic assignments and gene-copy number correction.

Click here for additional data file.

Table S2 Classification of abundant unaffiliated OTUs in the Cape Ferguson microbiomes

Classification performed using RDP (80% confidence) and Silva’s SINA (95% identity).

Click here for additional data file.

Table S3 Distance from the sampling sites to the nearest influencing rivers and river discharges during sampling weeks compared to seasonal averages

∗ Average for the sampling and six preceding days (7–13 Oct 2009 for Dunk Island, 8–14 May 2012 for Cape Ferguson). ∗∗ Average for the periods spanning 1 Apr–31 Oct in the years 2009–2012. ∗∗∗ Average for the periods spanning 1 Nov–31 Mar in the years 2009–2012. Data source: Queensland Government Department of Natural Resources and Mines Water Monitoring Information Portal ( http://water-monitoring.information.qld.gov.au).

Click here for additional data file.

Table S4 Maximum air temperatures (in °C) recorded at the sampling sites compared to their seasonal average

∗ Average for the sampling and six preceding days (7–13 Oct 2009 for Dunk Island, 8–14 May 2012 for Cape Ferguson). ∗∗ Average for the periods spanning 1 Apr–31 Oct in the years 2009–2012. ∗∗∗ Average for the periods spanning 1 Nov–31 Mar in the years 2009–2012. Data source: Australian Government Bureau of Meteorology Climate Data Online ( http://www.bom.gov.au/climate/data).

Click here for additional data file.

Table S5 Diuron and PSII HEq measurements near the Dunk Island and Cape Cleveland sites between 2005 and 2010

Data source: Lewis et al., 2009; Kennedy et al., 2012a; Kennedy et al., 2012b.

Click here for additional data file.

Table S6 Water parameters and quality index near the Dunk Island and Cape Ferguson sites for the sampling years

Each water quality index values is measured over four years. PN denotes particulate organic nitrogen, PP particulate phosphorus, Chl a chlorophyll a, and SS suspended solids. The water quality index was color-coded as: dark green (very good), light green (good), yellow (moderate), orange (poor), and red (very poor). Data source: Thompson et al., 2014.

Click here for additional data file.

Text S1 Meteorological and hydrological comparison of Dunk Island and Cape Ferguson

Click here for additional data file.

The authors would like to thank Elke Allers (the University of Arizona) for sample extraction assistance, Nicola Angel (Australian Centre for Ecogenomics) for her assistance with Illumina amplicon sequencing, and Jochen Müller (National Research Centre for Environmental Toxicology), Michael Nefedov and Nancy Lachner (Australian Centre for Ecogenomics) for assistance and advice in the lab.

Additional Information and Declarations

Competing Interests

Author Contributions

DNA Deposition

Data Availability

The authors declare there are no competing interests.

Florent E. Angly performed the experiments, analyzed the data, wrote the paper, prepared figures and/or tables, reviewed drafts of the paper.

Olga Pantos performed the experiments, wrote the paper, reviewed drafts of the paper.

Thomas C. Morgan performed the experiments, analyzed the data, reviewed drafts of the paper.

Virginia Rich and Philip Mercurio conceived and designed the experiments, performed the experiments, reviewed drafts of the paper.

Hemerson Tonin performed the experiments, reviewed drafts of the paper.

David G. Bourne, Andrew P. Negri and Gene W. Tyson conceived and designed the experiments, reviewed drafts of the paper.

The following information was supplied regarding the deposition of DNA sequences:

PRJNA276057

SRR1819825

The following information was supplied regarding data availability:

All raw data (DNA sequences and associated metadata) were submitted to GenBank.

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
