# Peer review of "Diuron tolerance and potential degradation by pelagic microbiomes in the Great Barrier Reef lagoon"

_PeerJ, doi:10.7717/peerj.1758_

## Round 0.1 · original submission · Major Revisions

As you can see from the reports of both expert reviewers, your study is considered to be of high interest. Having said that, there are some shortcomings, such as the lack of data on pollutant degradation, lack of abiotic controls in your experiments and the fact that different analyses have been done with different samples, and only provide snapshots of the situation.
Furthermore, conclusions are often too far reaching and strongly phrased, not backed up by the presented data.

Reviewer 1 ·

Basic reporting

All sequencing data sets should be uploaded to a relevant open-access database.

Experimental design

No comment

Validity of the findings

No comment

Additional comments

This manuscript examines the impact of diuron spiking on microbial community composition in the Great Barrier Reef. In field samples spiked in the lab with diuron and incubated for a year, the microbial community was investigated based on sequencing of a 16S rRNA fragment and the metagenome was analyzed for tolerance to diuron. Overall, the manuscript indicates that, at least under the experimental conditions presented here, spiking with diuron does not appear to have a significant impact on microbial community diversity or diuron tolerance. While this result has perhaps a lower impact than the authors may have anticipated, the fact that diuron spiking had such a minimal influence is in itself novel.

The manuscript is overall well written, in most cases very clearly stating and displaying results. My recommendations are below:

Was diuron measured during incubation? On the one hand, half-lives are given in the abstract; on the other hand there is no description of the sampling or measurement procedure in the materials and methods nor are there diuron degradation graphs. I would suggest including this information and making a more clear link between the diuron degradation that you apparently see either by photodegradation or biodegradation and the stability of microbial community structure. This is important to show that there was actually some activity in the incubation experiments. Additionally, I would be cautious in ascribing degradation to only biodegradation (as suggested in lines 328-350). As far as I can tell, there were no abiotic control experiments in which diuron was spiked and biological activity interrupted. Such a control is required to ensure that abiotic processes are not responsible for the transformations you observed.

Why was it chosen to do perform incubation experiments on the Cape Cleveland samples from 2012 and metagenome analysis of the Dunk Island samples from 2009? I could not find a scientific motivation for this. Can the authors give any indication of how similar the water composition and thus microbial community are on these temporal and spatial scales?

Based on Fig S3, it seems that the authors used high sequence depth but that richness and diversity were relatively low. Was this an unexpected result for the authors? Could the authors indicate to what extent this may be due to the low sampling volume (700 uL) used? Additionally, a table should be added giving for each sample, the number of reads, number of OTUs and the diversity and richness at the 20000 read cut-off.

The authors suggest that the samples are likely oligotrophic (line 261). Is there any data to support this claim? If the water comes from an area directly impacted by river water from an agricultural area, as implicated by the presence of diuron, then I would suspect that eutrophication may also be a relevant problem. Additionally, initial DOC concentrations and turnover during incubation would also be relevant, as these indicate the utilization of alternative substrates other than diuron. It would be good to include any information on water quality if possible.

Figure 2 in its current state is difficult to interpret. “Sampling day” is actually incubation time? Light presence is a nominal environmental factor? Thus, using an arrow, which implies a gradient in light quantities, inappropriate. The difference in circle sizes are difficult to read. I would suggest using a different shape for the different time points.

Reviewer 2 ·

Basic reporting

The paper needs at various points clarification (methodology, results, introduction, discussion)
Not all figures appear relevant, some seem double?
Some data of major importance seem to be missing like dynamics of diuron/34dichloroaniline concentrations in the microcosm

Experimental design

The experimental design misses abiotic controls for appropriate evaluation of diuron degradation
Metagenomic sequence analysis is performed on a sample different from the microcosm
Methodology should be better described for reproducibility

Validity of the findings

Statistics are not reported
Conclusions are sometimes too strong/speculative and should be reconsidered. Speculation is not identified as such.

Additional comments

This paper describes an interesting study on the effect of diuron and its degradation in marine pelagic microbial communities of the Great Barrier Reef lagoon in a controlled lab microcosm experiment. To the best of my knowledge, it is the first report on microbial communities interacting with pesticide in marine ecosystems. The authors use an up to date approach based on metagenomics analysis to study this and use relevant diuron concentrations which is appreciated. The authors do not find indications for effects on microbial community structure and apparently find minor removal of diuron. The authors try to explain this by examining the metagenome sequences looking at potential mutations in the psbA gene and the occurrence of known genes involved in diuron degradation.
Although the paper studies a very interesting question and presents a lot of data, it has several flaws which should be taken into account to deserve publication.
1. The authors hypothesize that changes in community composition might be due to the proliferation of organisms using diuron as a nutrient source. However, this is difficult to expect taking into account the trace level concentrations of diuron that were amended. Moreover, finally, diuron was hardly degraded.
2. The authors make use of batch microcosm systems in which the community was contained for about a year without refreshment of the medium and hence input of nutrients. As such, the community and the organisms within were completely depended regarding nutrient delivery of recycling and survival/growth from probably cells which died during the incubation. Moreover, invasion of potential diuron degraders could not take place. This should be at least take into account into the discussion.
3. The metagenomics data were from a snapshot sample taken in the lagoon at a different location from this were the sample was taken used in the microcosm experiment and as such the sequence data might be irrelevant for making conclusions about the observations done in the microcosm experiment. Do the authors have any idea how the sampling position affects the community composition?
4. The authors make relatively strong statements based on the metagenomics sequence data regarding the occurrence of potential diuron hydrolases while this remains very speculative without actual proof by performing protein expression experiments using the metagenomics sequence after recovery of the genes from the metagenome. The authors should be more humble in that.
5. The authors do not include abiotic controls in their experimental design to control abiotic losses of diuron. That is important to evaluate “degradation”. However, they detect 34dichloroaniline, a known metabolite but remain vague about this. The authors are quite vague regarding diuron concentrations in time in general.
6. Statistical analysis is lacking (of 16S rRNA gene abundances and diuron concentrations).

Specific remarks.
Lines 21-44: The abstract is quite long and contains much information that is not needed. The introductory part of the abstract is particularly long and when explaining the results, information about methodology can be often omitted and sentences can be shortened to the essence.
Line 62: change “found” into “detected”
Line 65: “water quality guidelines”. Can the authors be more specific about this. Threshold concentrations of diuron implying bad water quality?
Line 64: “wide range of marine organisms”: does it concern eukaryotic organisms? (see also next remark).
Lines 65-72 + line 73: When reading line 73, it appears that lines 65-72 only concerns eucaryotes bit that does not seem the case since the authors mention in line 67 benthic biofilms. Reading the title of the Magnusson paper, it appears it concerns biofilms of microalgae. This should be specified.
Line 70-72: Is this (accumulation) of relevance in a marine environment? I suppose the compound is directly diluted massively when it enters the marine environment. Is there any knowledge about accumulation in the sediment or of bioaccumulation/biomagnification of diuron?
Line 74-75: Please specify the concentration levels that gave these effects.
Lines 81-83: Is this an alternative explanation? Concentrations of diuron in such systems will be too low to induce extensive growth unless in these studies used concentrations were relatively high (but probably irrelevant).
Lines 88-89: Can be removed from the paper.
Line 93-94: Concentrations will be too low for observable growth unless the organisms were initially present in extremely low densities. The authors could easily calculate theoretical yields in case diuron is used as carbon source.
Line 95: Please specify that also diuron removal was monitored.
Line 103: Specify reported concentrations of diuron.
Line 105-107: Did the authors include an abiotic control (see general remarks)?
Line 110: How were the samples from the flasks taken? Were the flasks shaken for homogeneity before sampling?
Line 111-114: This the authors look at life and dead cells for instance by using appropriate staining kits. It would have been of interest that the authors report somewhere the results of this microscopic analysis, i.e., dynamics of cell numbers in time.
Line 115: You cannot extract DNA by simple centrifugation. Please rephrase.
Lines 119-120: This sentence is unclear.
Line 121: What is a “microbial amplicon”?
Line 121: Does the same primer set amplify as well archaea as Bacteria 16S rRNA genes? This should be clarified. Is the Dove et al (2013) reference the right one? This is not apparent based on the title of the publication which does not seem to deal with microbial community analysis.
Line 195: Why metal dependent hydrolase superfamily? Do PuhAB belong to this family? Please specify.
Line 201: And the alignment with other hydrolases?
Line 230: “diuron incubation experiment” This sounds like lab language. Please rephrase.
Line 231: “4.96 million pairs of reads”. Specify it is collectively over the 72 samples.
Lines 230-235: How was dealt with the sequence data in case the number of reads were different between the different samples?
Line 240-250: Did the authors used statistics to underpin their data? Were these differences significant based on the replicate data. Please specify and provide significance levels.
Figure 1 caption: “diuron experiment” this is again lab language.
Lines 263-264: again what was the variation regarding diuron concentration between replicates. Was the difference between dark and light conditions significant? It would be welcome to include a figure showing the diuron concentration dynamics in time.
Lines 292-294: The relationship between mutations in the PsbA protein and diuron and other herbicides should be specified. What did these reference papers actually found and reported regarding this item? Now it is quite vague. Reading this paragraph like it is, it seems that the role of PsbA adaptation in diuron tolerance is not yet elucidated and hence still speculative. This means that even when PsbA variants were found in the metagenomics sequences it remained speculative that these explained the observed non-effect of diuron on the community.
Lines 295-304: Why not done on a sample from the microcosm experiment itself? Moreover, since there is such a high dynamic of community composition in the microcosm!
Line 301: “to confer resistance” To which compound? This should be specified and better related to the information (which should be improved) provided in lines 290-294.
Lines 304-313: The rationale to include this paragraph is not clear. Is this related to a mutation to adapt to irgarol?
Lines 314-320: 319-320: It is not clear how this can contribute to diuron tolerance?
Lines 325-326: Were these differences significant? How can degradation be evaluated without abiotic control?
Line 326: What does the Mercurio et al paper actually tell us? Does this paper reports on the diuron concentration data? The description of this reference in the reference list is unclear. Better to include these data in the current paper.
Line 327-328: detection of 34dicholroaniline. This is very important and should be reported in the abstract. However, its detection should be specified. In which flasks? At which time point? Stoichiometric production with diuron removal?
Line 330: do you mean number of heterotrophs or composition?
Line 331-333: This sentence needs rephrasing.
Line 340-341: See general remarks about concentration effect on growth. Certainly when diuron is hardly degraded.
Line 351: “likely”. This is quite a strong statement (see general remarks). Highly speculative instead.
Line 359: “marine microbial seed bank” This is unclear.
Line 364-366: “as seen in …” This is unclear. Please rephrase and clarify.
Line 367: “high” is relative. Please provide information about concentrations.
Line 373: “herbicide research” sounds like lab language (rephrase)

Figure S2 seems not needed
Figure S4: difference with figure 1???
Figure S3: variation between replicates should be indicated.

---

## Round 0.2 · accepted · Accept

You certainly did an excellent job in addressing all issues raised by the reviewers.